# CRISPR/cas9 Allows for the Quick Improvement of Tomato Firmness Breeding

**DOI:** 10.3390/cimb47010009

**Published:** 2024-12-29

**Authors:** Qihong Yang, Liangyu Cai, Mila Wang, Guiyun Gan, Weiliu Li, Wenjia Li, Yaqin Jiang, Qi Yuan, Chunchun Qin, Chuying Yu, Yikui Wang

**Affiliations:** 1Vegetable Research Institute, Guangxi Academy of Agricultural Sciences, Nanning 530007, China; qhyang90@163.com (Q.Y.); liangyuc5581@163.com (L.C.); milawang0604@163.com (M.W.); ggyun19@163.com (G.G.); liweiliu@gxaas.net (W.L.); lwj3386@gxaas.net (W.L.); jiangyaqin@gxaas.net (Y.J.); yqyuan12@163.com (Q.Y.); qinchunchun2024@163.com (C.Q.); 2College of Agriculture, Guangxi University, Nanning 530004, China

**Keywords:** tomato, CRISPR/cas9, fruit firmness, germplasm improvement, transgenic-free

## Abstract

Fruit firmness is crucial for storability, making cultivating varieties with higher firmness a key target in tomato breeding. In recent years, tomato varieties primarily rely on hybridizing ripening mutants to produce F_1_ hybrids to enhance firmness. However, the undesirable traits introduced by these mutants often lead to a decline in the quality of the varieties. CRISPR/Cas9 has emerged as a crucial tool in accelerating plant breeding and improving specific target traits as technology iterates. In this study, we used a CRISPR/Cas9 system to simultaneously knock out two genes, *FIS1* and *PL*, which negatively regulate firmness in tomato. We generated single and double gene knockout mutants utilizing the tomato genetic transformation system. The fruit firmness of all knockout mutants exhibited a significant enhancement, with the most pronounced improvement observed in the double mutant. Furthermore, we assessed other quality-related traits of the mutants; our results indicated that the fruit quality characteristics of the gene-edited lines remained statistically comparable to those of the wild type. This approach enabled us to create transgenic-free mutants with diverse genotypes across fewer generations, facilitating rapid improvements in tomato firmness. This study offers significant insights into molecular design breeding strategies for tomato.

## 1. Introduction

Tomato is one of the Solanaceae family’s most widely cultivated vegetable crops globally. In 2024, global tomato production is projected to reach 47 million tons, solidifying its status as one of the most significant vegetables worldwide (FAO, 2024). The primary part of the tomato that humans consume is its ripe fruit. This characteristic renders tomato an excellent option for a healthy diet, attributable to its high levels of vitamin C, potassium, and antioxidants [1,2]. The tomato originated in South America and has evolved to exhibit larger fruit sizes and greater diversity over thousands of years of human cultivation [3,4]. In their natural state, tomatoes soften as they ripen, leading to a shortened shelf life and reduced commercial value [5,6]. Consequently, breeding tomato varieties with an extended shelf life has recently become a primary focus [7]. Investigating the mechanisms that regulate fruit firmness is essential, as it not only enhances the commercial value of tomato, but also mitigates post-harvest damage and waste [7].

Fruit firmness is intrinsically linked to the ripening process. As a quintessential climacteric fruit, tomato is often bred using late-maturing varieties as hybrid parents to enhance fruit firmness and storability [8]. A notable example is the *ripening inhibitor (rin)* mutant, widely used to fortify tomato fruit firmness. This mutation is characterized by its inability to facilitate normal maturation, failing to produce lycopene, a lack of fruit softening, and an absence of the ethylene burst typically associated with ripening [9]. The F_1_ obtained by using *rin* mutant as a hybrid parent has significantly improved firmness and storability [10].

Previous studies indicate that *rin* represents a functionally acquired mutant [11]. RIN is an essential transcription factor that interacts with ethylene to modulate gene expression in the fruit cell wall metabolic pathways [12]. This interaction alters the cellular composition, producing fruit softening during ripening [9,11,12]. However, the limitations of utilizing the *rin* mutant are becoming increasingly apparent. The absence of *RIN* results in the inability of tomato to undergo normal ripening processes. Although exogenous ethylene can induce a color change to red, it does not promote complete ripening, degrading the overall fruit quality [11]. Furthermore, mutations in other genes, such as *NOR*, *CNR*, and *TAGL1*, also disrupt normal tomato ripening, thereby constraining their use in breeding [10]. Therefore, exploring innovative strategies to enhance tomato firmness without compromising quality is imperative.

The changes and degradation in cell components, particularly cell wall constituents, can directly reduce the structural strength of fruits [13,14]. The cell wall comprises components such as cellulose, hemicellulose, and pectin, where the stability of the pectin molecular main chain or side chains is closely related to fruit strength [15]. The depolymerization, dissolution, and reduction in pectin polymer molecules are direct causes of fruit softening during the maturation process [16,17]. The depolymerization and dissolution of pectin are regulated by a series of enzymes, including polygalacturonase (PG), β-galactosidase (β-gal), pectin methylesterase (PME), xyloglucan endotransglucosylase/hydrolase (XTH), and pectin lyase (PL) [18]. Pectin lyase, encoded by the *PL*, can cleave the α-1,4-glycosidic chains, thereby facilitating the degradation of pectin macromolecules into smaller molecules, ultimately resulting in the fruit’s softening [19]. Inhibiting the depolymerization and dissolution of pectin can effectively delay fruit softening. For instance, utilizing RNA interference (RNAi) technology to suppress the expression of *PG* or *PL* can significantly increase fruit firmness and extend its shelf life [19,20,21,22,23]. However, due to various controversies and restrictions surrounding transgenic technology in recent years, the genes that have the potential to improve tomato firmness have not been effectively utilized [24].

Another factor influencing the firmness of tomato is the thickness of the cuticle. The cuticle consists of cutin and wax, forming a complex of hydrophobic compounds with various components, contents, and structures [25,26]. It serves as a mechanical barrier for the outer layer of the fruit and is closely related to the fruit’s storability, quality, and abiotic stress resistance [27]. Increasing the content of surface wax or cutin in tomato fruits through genetic engineering can significantly enhance fruit firmness and improve storability [23]. *FIS1* encodes a GA2-oxidase, which plays an important role in the biosynthesis of tomato wax and cuticle [25]. The loss-of-function mutation in *FIS1* can improve the biosynthesis of cutin and wax, thereby increasing fruit firmness and storability, with no adverse effects on fruit weight and quality [25]. Therefore, the gene regulation of fruit firmness and storability through the synthesis pathways of wax and cutin provides potential strategies and references for the molecular breeding of tomato.

In recent years, the rapid development of CRISPR/Cas9 technology and the cloning of key genes for important agronomic traits have made the efficient and precise molecular breeding of tomato a reality. Significant progress has been made in improving tomato quality, increasing yield, and enhancing stress resistance through gene editing technology [28,29,30,31]. To enhance the firmness of tomatoes, we selected two dominant genes from distinct pathways for gene editing. In this study, we utilized the CRISPR/Cas9 system to knock out the *FIS1* and *PL* that regulate fruit firmness in cultivated tomato varieties OT44 and OT45. As a result, the gene-edited lines with different genotypic mutations were obtained, from which high-quality tomato mutants with improved firmness and storability were screened. Our study aims to investigate a new pathway for the rapid improvement of tomato fruit firmness across fewer generations, while also providing a reference for enhancing other traits.

## 2. Materials and Methods

### 2.1. Material and Cultivation Conditions

OT44 and OT45 are varieties of large-fruited tomato, with fruit colors of pink and red, respectively, and a relatively soft texture when ripe. The fruit weight of OT44 is approximately 70 g, with a soluble solid content of 9% and a firmness of 40N at maturity. In contrast, the fruit weight of OT45 is 85 g, featuring a soluble solid content of 7% and a firmness of 84N at maturity. Both tomato varieties exhibit good flavor quality; however, they share the drawback of becoming soft post maturity. Consequently, we selected these two tomatoes as candidates for enhancing firmness. They were developed through multiple generations of self-pollination by the Vegetable Research Institute of the Guangxi Academy of Agricultural Sciences. All plant materials utilized in this experiment were cultivated in a greenhouse, adhering to two annual planting cycles under long-day conditions (16 h of light and 8 h of darkness) at a temperature range of 26–30 °C.

### 2.2. Vector Construction

We selected the first exon of *FIS1* and the second exon of *PL* to design target sequences aimed at generating frameshift mutations or large deletions to the greatest extent possible. Four targeting sequences were designed using the CRISPR-P tool (http://cbi.hzau.edu.cn/cgi-bin/CRISPR (accessed on 20 October 2022)) to construct the CRISPR/Cas9 multi-target binary vector. Four sgRNAs were synthesized through PCR reactions and separated by tRNA. Using the Golden Gate assembly method, they were cloned into the BsaI restriction site of the pTX 041 vector. The specific methods are referred to in previous studies [31,32].

### 2.3. Tomato Genetic Transformation

To obtain transgenic plants, a method of *Agrobacterium*-mediated genetic transformation was used to introduce a binary vector containing four sgRNAs into OT44 and OT45 [33]. The presence of T-DNA was confirmed in T_0_-positive plants by PCR detection of specific T-DNA fragments. Subsequently, T_1_ generation plants were obtained through self-pollination, and the same detection method was used to screen for T-DNA-free lines. The forward primer is designated as Cas9-Fw, while the reverse primer is referred to as Cas9-Rv. The primer sequences are shown in Appendix A.

### 2.4. Target Editing Efficiency Detection and Genotyping

To assess the editing efficiency of T_0_ transgenic plants, we designed detection primers on both sides of each target sequence, as shown in Figure 1a. Using the T_0_ transgenic plants as templates, we performed PCR amplification of specific fragments. The PCR products were then ligated into the pEASY-Blunt vector and transformed into *Escherichia coli* to obtain clones following the guidelines provided in the pEASY^®^-Blunt Cloning Kit (TransGen, Beijing, China, CB101-01). These were subjected to Sanger sequencing to determine the sequence mutation status. Based on the genotyping results, we screened for single and double mutants for further analysis. The primer sequences are shown in Appendix A.

### 2.5. Phenotypic Identification and Statistical Analysis

T-DNA-free single plants with mutations in the target gene were selected and subjected to phenotypic analysis. Mechanical stress tests were conducted on tomato fruits at the red ripening stage using a texture analyzer (Brookfield CT3) with a plate compression method [25]. The firmness of the tomato fruit was measured by the maximum pressure exerted at the time of fruit rupture [25]. Soluble sugar content was measured using a portable refractometer (ATAGO, PAL-1). For each line, three plants were measured, with at least five tomato fruits at the red ripening stage assessed per plant; the average value for each group served as the line’s fruit soluble sugar content. The fruit ripening time was recorded from the flowering development of the tomato fruit to various stages of maturity, with three fruits from each line sampled at three different time points for statistical analysis. Fruit size was measured by recording both the horizontal and vertical diameters of the fruit from development to the red ripening stage, taking measurements from three fruits per line and calculating the average. The weight of five red ripening fruits from each line was measured using an electronic scale, and the average fruit weight was calculated. Each line was repeated three times. Additionally, five red tomato from each line were stored at room temperature for four weeks, with the fruit weight measured weekly; the rate of weight loss served as an indicator of fruit dehydration. Significant differences were calculated using the Tukey–Kramer test, with different letters indicating a highly significant difference (*p* < 0.05).

### 2.6. The Methodologies for Cytological Observation

The mature green tomato pericarp was sliced into pieces measuring approximately 1.5 cm in length and 0.5 cm in width. Subsequently, the samples were stained with a 0.05% aniline blue solution for 2 min, then rinsed with water for 5 min. The samples were observed and photographed under a light microscope. Image J software was used to analyze the images and statistically assess the pericarp cells’ size and layers. Paraffin sections of mature green and red ripening tomato pericarp were prepared. After Sudan IV staining, changes in the cuticle were observed under a microscope following methods referenced in previous studies [34].

## 3. Results

### 3.1. Editing Results and Efficiency

To obtain the *FIS1*, *PL* single mutants, and double mutants, we utilized an Agrobacterium-mediated genetic transformation system in tomato to introduce a CRISPR/Cas9 multi-target binary vector containing four target sgRNAs (Figure 1a) into the elite inbred lines OT44 and OT45. Eighty-six positive T_0_ transgenic plants were obtained in the OT44, and 58 positive T_0_ transgenic plants were obtained in the OT45. Through PCR amplification and Sanger sequencing, we confirmed that the target genes in the positive plants had been edited to varying extents with different editing efficiency (Table 1). By self-pollination and genotyping, we obtained edited progeny with different genotypes, including 13 *FIS1* single mutants, 9 *PL* single mutants, and 9 double mutants in the OT44 background, which were named CR-*fis1*-OT44, CR-*pl*-OT44, and CR-*fis1/pl*-OT44, respectively. In the OT45 background, we obtained 6 *FIS1* single mutants, 6 *PL* single mutants, and 8 double mutants, named CR-*fis1*-OT45, CR-*pl*-OT45, and CR-*fis1/pl*-OT45, respectively. These editing lines encompass deletion, insertion, and substitution mutations. The mutation status of the target sites is shown in Figure 1b and Appendix A. Given that our target sites are situated within the first or second exons of the two genes, an insertion or deletion of a single base can induce a frameshift mutation, ultimately leading to the premature termination of protein translation. We selected editing lines that resulted in frameshift mutations resulting in loss of gene function for further investigation.

### 3.2. Genotype and Phenotype of Mutants

To identify the phenotypic variations caused by different gene mutations, we selected single and double mutants without exogenous genes to examine the firmness of their fruit at maturity. The results indicated that the mechanical stress resistance of the fruit in CR-fis1, CR-pl single mutants, and double mutants was significantly enhanced compared to their background materials, with the compression resistance of the double mutants being notably higher than that of the single mutants (Figure 1c). Additionally, we found that, in the OT44 background, the compression resistance of the fruit in the CR-fis1 single mutant was higher than that in the CR-pl single mutant, whereas this difference was not observed in the OT45 background (Figure 1c).

We tested the water loss rate of red ripening fruits from various genotypes to further investigate the relationship between fruit firmness and shelf life. Under conditions of 24 °C, the water loss rate of the wild-type materials was significantly higher than that of the mutant materials, while the double mutant materials exhibited the lowest water loss rate (Figure 2b). After 2 weeks of treatment, the epidermis of the wild-type fruit showed signs of shriveling, which was subsequently followed by fungal infection and decay, further shortening the shelf life of the fruit (Figure 2a). Meanwhile, we found that the CR-fis1 lines were more susceptible to fungal infection compared to the CR-pl lines. In the OT44 background, the water loss rate of the CR-fis1 lines was lower than that of the CR-pl lines after 2 weeks, whereas, in the OT45 background, there was no significant difference in the water loss rates between the two lines.

### 3.3. Cytological Observation

We utilized microscopy to examine and quantify the mature green fruit of both wild-type and mutant specimens. We aimed to investigate the cytological basis underlying the enhanced firmness and post-harvest storability observed in the mutants. We found no significant differences in the thickness of the pericarp, cell shape, and cell arrangement between the CR-*fis1* mutant line and the wild type. However, the number of cell layers in the pericarp of the CR-pl mutant and double mutant significantly increased compared to the wild-type fruit, while the cell area significantly decreased (Figure 3).

The firmness of tomato fruit is also influenced by the thickness of the cuticle on the surface. To further investigate the reasons for the increase in firmness and storability of the mutant fruit, we selected wild-type and mutant fruit at the mature green (MG) stage and red ripening (RR) stage to prepare paraffin sections to observe their cuticle structure. The results indicated that the cuticle thickness of CR-*fis1* and CR-*fis1/pl* mutant fruits was significantly thicker compared to that of the wild type, while there was no significant change in the cuticle of CR-*pl* mutant fruits compared to the wild type (Figure 4).

### 3.4. Other Agronomic Traits

We analyzed the fruit maturation period, shape, size, and soluble solids to determine whether the mutations in the *FIS1* and *PL* affect other fruit traits while enhancing the firmness of tomato fruits. The results showed that, compared to the wild-type fruits, there were no significant changes in the ripening period, fruit shape, size, or the soluble solids in the mutants (Figure 5).

## 4. Discussion

The domestication of tomato by humans has focused not only on enhancing yield, but also on improving quality. This quality enhancement encompasses several dimensions, including increasing nutritional content, aesthetic appeal, and prolonging storage life [35]. Among the various factors influencing quality, increasing fruit firmness is particularly effective for enhancing storability. Additionally, greater fruit firmness is beneficial in minimizing losses during harvesting and transportation, thereby significantly contributing to cost savings and improved economic returns. Traditional breeding methods typically involve using F_1_ hybrids derived from crossing mature mutants to enhance fruit firmness and extend shelf life. However, this conventional approach is time-consuming, often requiring several years of extensive field evaluations, and inevitably introduces adverse effects associated with integrating mature mutants [36]. Consequently, there has been a persistent demand for more efficient and precise crop improvement techniques. In this study, we successfully developed various tomato varieties exhibiting enhanced firmness and other desirable traits by precisely applying gene editing technology. This advancement holds significant potential for expediting the breeding process of tomato.

Ripening mutants commonly utilized in traditional breeding, such as the *rin* mutant, are known to enhance fruit firmness and extend shelf life by suppressing pectin decomposition. However, these mutants exhibit notable limitations, including an inability to accumulate lycopene and suboptimal flavor profiles [37]. Previous studies have demonstrated that *RIN* functions as a critical ripening regulator, playing an essential role in the dissolution of cell walls in fruits [11]. RIN regulates fruit softening by modulating the expression of key enzyme genes involved in pectin synthesis and degradation, including CEL2, XYL1, EXP1, PL, PG, and TBG4 [9,11,12].

The limited applicability of transcription factors due to their pleiotropic effects is also observed in mutants such as *nor*, *cnr*, and *tagl1* [8,23,38]. Consequently, employing genetic engineering to inhibit the expression of key enzymes has emerged as a more advantageous strategy for enhancing tomato firmness. For instance, the use of RNAi technology to suppress the expression of *PG* and *PL* has been shown to improve fruit firmness and storability significantly [19]. Nevertheless, transgenic approaches, including RNAi, face numerous challenges related to ethical considerations, legal regulations, and consumer acceptance, which restrict their practical applications [24]. Therefore, CRISPR/Cas9 technology, capable of efficiently editing target genes without introducing exogenous genes or undesirable traits, has emerged as a preferred method in genetic modification [39]. In our study, we created several mutants without any exogenous genes. The results indicated that the mechanical stress resistance of the fruit in those mutants was significantly enhanced compared to their background materials (Figure 2a). Our findings indicate that editing either *FIS1* or *PL* individually can enhance fruit firmness, with comparable improvements observed in OT45. However, the lines edited for *FIS1* demonstrate less enhancement than those edited for *PL* in OT44. This observation suggests the presence of additional factors influencing the effect of the *PL* gene in OT44, warranting further investigation utilizing a broader range of genetic background materials.

CRISPR/Cas9 technology has become pivotal in expediting plant breeding and enhancing specific target traits. Numerous studies conducted in recent years have demonstrated its considerable application potential [28,40,41]. Pectin lyase, encoded by the *PL*, can cleave the α-1,4-glycosidic chains, thereby facilitating the degradation of pectin macromolecules into smaller molecules, ultimately resulting in the fruit’s softening. Notably, the inhibition of *PL* expression has increased fruit firmness without causing significant changes in flavor components such as sugar and acid content [18]. In our study, we knocked out the *PL* via the CRISPR/Cas9 system, and the outcomes aligned with previously observed effects associated with the silencing of *PL* expression via RNAi [18]. Furthermore, the ability to screen for plants devoid of transgenic components in the edited progeny will enhance the applicability of CRISPR-mediated mutants in breeding [39]. *FIS1* encodes a GA2-oxidase, and mutations in *fis1* lead to a cuticle that is thicker than wild-type plants [25]. This increased cuticle thickness contributes to greater external support for the fruit, thereby augmenting its firmness while maintaining yield and quality [25]. Our study demonstrates that there is no significant difference in storage duration between the *CR-fis1* and CR-*pl* mutants. However, when assessing the rate of water loss, we observed that *CR-fis1* mutants were more effective at maintaining their intact appearance compared to CR-*pl* mutants. This observation may be attributed to the fact that *CR-fis1* mutants possess a thicker cuticle. Therefore, the simultaneous editing of these two genes can significantly enhance the firmness of tomato, thereby ensuring a higher-quality shelf life. This approach not only extends storage duration, but also preserves the better appearance of tomato.

Based on a synthesis of prior research, two primary approaches have been identified for enhancing tomato firmness: the first involves regulating fruit softening by modulating the strength of pulp cells, while the second focuses on improving fruit mechanical strength by increasing cuticle thickness [19,25]. This study focused on the *PL* and *FIS1* for gene editing, leveraging these distinct mechanisms. Our findings indicate that the effects of concurrent mutations in both genes were significantly stronger than those achieved through individual gene mutations. Moreover, we observed a notable increase in the number of cortical cells in the CR-*pl* mutant, alongside a decrease in cell area (Figure 3). This suggests that *PL* plays a role in regulating cell wall degradation and influences fruit strength by altering the mode of cell division. In the future, as an increasing number of significant agronomic-trait-regulating genes are identified, the pursuit of precise molecular design breeding for tomato through gene editing is expected to become standard practice.

## 5. Conclusions

The application of gene editing technology to simultaneously modify regulatory genes across different mechanisms can facilitate a multi-channel superposition effect and enhance the efficiency of germplasm improvement. By precisely editing genes within two distinct regulatory pathways, we successfully generated new germplasm exhibiting varying levels of firmness in tomato. Furthermore, the simultaneous editing of two genes facilitated the enhancement of tomato firmness without compromising other traits. Despite the current limitations of research on CRISPR/Cas9 in this article, which is primarily focused on gene knockout, this advancement significantly extends the shelf life of tomatoes and improves breeding efficiency. Our study provides a new pathway for efficient genetic improvement and modern tomato breeding.

## Figures and Tables

**Figure 1 cimb-47-00009-f001:**
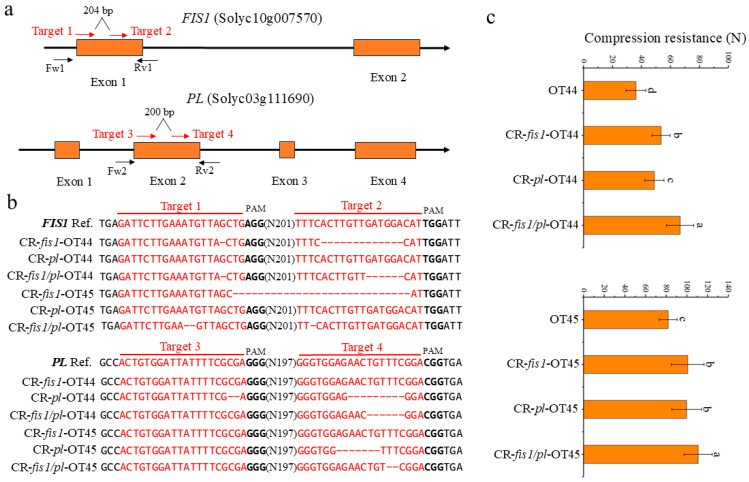
Rapid improvement of tomato firmness using the CRISPR/Cas9 technology. (**a**) The schematic showing the gene structure of *FIS1* and *PL*; four target sgRNAs were designed for gene edit. The red arrows represent the four target sgRNAs, and the black arrows represent the corresponding detection primers. (**b**) The detail sequences of gene editing lines. (**c**) The compression resistance of wild-type and mutant fruits. Significant differences are calculated using the Tukey–Kramer test, with different letters indicating a highly significant difference (*p* < 0.05). Error bars, mean ± SD. *n* = three biological replicates.

**Figure 2 cimb-47-00009-f002:**
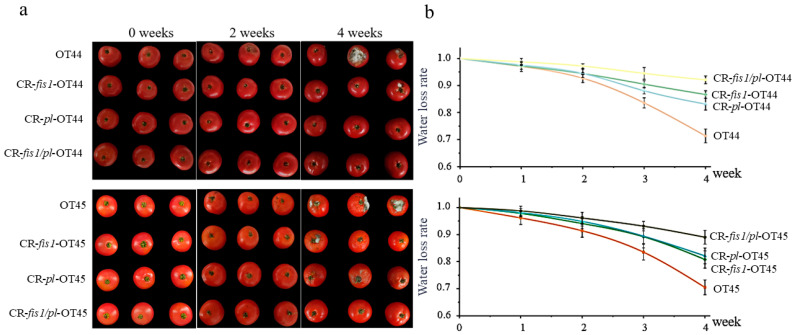
Appearance changes and water loss rate of wild-type and mutant fruits during storage. (**a**) Appearance changes in wild-type and mutant fruits during storage at the red ripening stage. (**b**) Water loss rate of wild-type and mutant tomato fruits.

**Figure 3 cimb-47-00009-f003:**
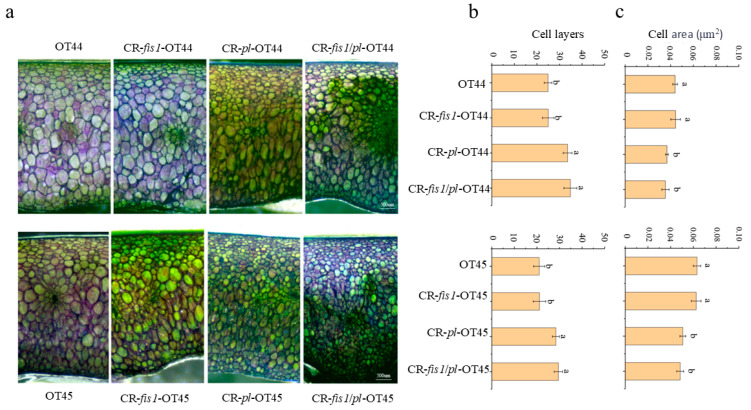
Microscopic observation of pericarp in wild type and mutant. (**a**) Comparison of pericarp thickness. (**b**) Observation of pericarp cells, scale bar = 500 μm. (**c**) The cell area of pericarp in wild type and mutant. Significant differences are calculated using the Tukey–Kramer test, with different letters indicating a highly significant difference (*p* < 0.05). Error bars, mean ± SD. *n* = three biological replicates.

**Figure 4 cimb-47-00009-f004:**
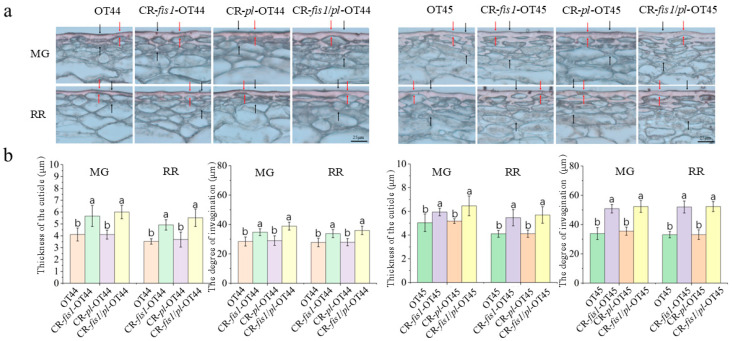
Thickness of the cuticle on the epidermis in wild type and mutants. (**a**) Cuticle sections stained with Sudan IV to visualize the cutinization of epidermal cell walls. (**b**) The thickness of cuticle and the degree of invagination. Red arrows indicate the degree of invagination of the cutinization, while black arrows indicate the thickness of the cuticular layer. Scale bar = 25 μm. Significant differences are calculated using the Tukey–Kramer test, with different letters indicating a highly significant difference (*p* < 0.05). Error bars, mean ± SD. *n* = three biological replicates.

**Figure 5 cimb-47-00009-f005:**
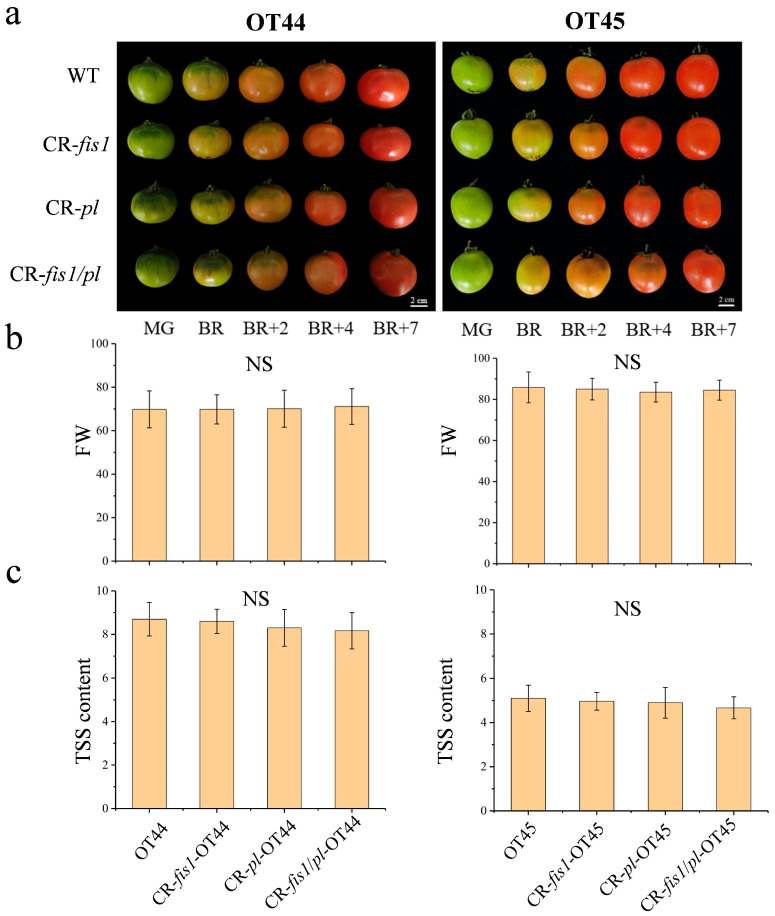
The appearance and quality of fruits in wild type and mutants. (**a**) The appearance of fruits in wild type and mutants. (**b**) The weight of wild-type and mutant fruit. (**c**) The soluble solids content of wild-type and mutant fruit. FW, fruit weight. TSS, soluble solids. Significant differences are calculated using the Tukey–Kramer test (*p* < 0.05). Error bars, mean ± SD. n = three biological replicates. NS, no significance. Scale bar = 2 cm.

**Table 1 cimb-47-00009-t001:** The editing efficiency of the transgenic plants.

Background	Mutant	Number in T_1_ Generation	Editing Efficiency (%)
OT44	CR-*fis1*	13	41.9
	CR-*pl*	9	34.6
	CR-*fis1/pl*	3	31
OT45	CR-*fis1*	6	40
	CR-*pl*	6	35.3
	CR-*fis1/pl*	8	30.8

## Data Availability

All datasets generated in this study are included in the article and Appendix A.

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
