# Peer review of "CRISPR/cas9 Allows for the Quick Improvement of Tomato Firmness Breeding"

_cimb, 2024, doi:10.3390/cimb47010009_

Round 1

Reviewer 1 Report

Comments and Suggestions for Authors

Please see review report  

Author Response

Comments for authors 

- Thank you so much for your work. There are some required improvements to improve this manuscript.

Response:Thank you for your thorough and insightful comments. In light of your comment, we have meticulously revised the manuscript. Below, please find our point-by-point responses to your remarks.

Abstract 

- This section should contain an introduction, material and methods, and results

 Response: Thanks for your good advice, the abstract has been revised.

Introduction

- The authors should write something about the importance of tomato 

Response:Agree. We have added the relevant content to the article, as indicated on line 110.

- From 28-33 without references??? In scientific publications, the authors should not write any phrases without references.

Response:The references have been added.

- The phrase “Our study provides a new pathway for efficient genetic improvement and modern tomato breeding” should be in the conclusion section.

 Response:Thank you for pointing this out. We have modified the corresponding contents as required. - What is the main aim of the study??? It is not mentioned usually at the end of the introduction section.

Response:The main aim has been added in lines 94-96.

Material and methods

- in this section, the authors do not use any references, in the scientific publication when we write any parts in the manuscripts, we should cite it. 

- 89-94; 96-101; 103-107; 109-115; 117-134; 136-141 all of these paragraphs without any references why???

Response:Thank you for bringing this to our attention. The descriptive content of the method presented in our study, specifically from lines 89-94, 96-101, 123-134 and 134-141, is original and not sourced from existing literature; hence, no citations are included in this section. To facilitate a better understanding for readers, we have added detailed operational steps. Lines 103-107 as shown in the text cite reference [40]. Lines 109-115, we followed the methods provided in the pEASY®-Blunt Cloning Kit (TransGen, CB101-01). Lines 117-122, we cited the reference [29].

- What about statistical analysis???? The authors should not mention anything about statistical analysis??

Response:Thank you for pointing this out. The statistical analysis was present in section 2.5

Results  

- Figures 1 and 2 (a and b) should be made more visible. They are not good;

please try to improve them.

Figures 3, 4 and 5 are not good, please improve them. 

Response:The images in Fig. 1-5 have been upgraded to higher quality versions.

Discussion 

- The paragraph (275-284) without any references, please link this paragraph with your results.

  Response:  The reference has been added.

Conclusion

- The conclusion needs to be improved, and the authors should focus on the most important results and their applications in the agriculture sector and to the farmers with their income. 

- The authors should focus on their future vision and their further study.  

Response:Thank you for your thoughtful comment. We have revised this section in accordance with your suggestions.

References 

- There are many references that are very old and the authors should update them and replace them with the most recent references (it is preferable to be at the last 5 years old) 

- The names of authors should summarize the names of journals.

Response:We have updated the cited literature in our manuscript. Some of the referenced literature is dated, as the methodologies we cite have not undergone significant updates, such as references [17-23].

Reviewer 2 Report

Comments and Suggestions for Authors

In the manuscript named “CRISPR/cas9 allows for the quick improvement of tomato firmness breeding”, authors have adopted CRISPR/cas9 technology to knock out the FIS1 and PL in tomato, their results have shown the gene editing lines with improvement of firmness in tomato. These findings would be useful for tomato genetic breeding in future, but there were some comments about it.

(1) Authors have knocked two genes, but their results have not shown the detail about gene editing lines, including DNA sequencing results. The representational sequences should be displayed in in manuscript, and other results should be released by supplements, or other public method.

(2) How these two genes express in gene editing lines? If there are gene expressing results present in this research, such as qRT-PCR, which would be beneficial for exploring genes’ function, and trait formation.

(3) The results of gene editing were unclearly described, which effects were these gene editing sites? Unknown, how did these editing sites function or effect on enzyme function, authors would provide more evidences about their function.

(4) The genetic background about two tomato lines, OT44 and OT45, were not well introduced in this research, why selected these two lines? They were different between some traits, such as compression resistance, soluble solids, how did two genes, FIS1 and PL, function about these traits, please describe in detail.

(5) The refs were missed in many sections, such as method section, many words were selected from public methods, but refs missed. The same as discussion section, few refs were listed, and discussion needed to be described in depth.

(6) The conclusion was rather superficial and fails to be integrated with present research.

(7) There was an excess space in line 155.

(8) Fig. S1 could not be accessed in present version, please check it.

(9) The statistical analysis was not clearly described, ANOVA, or other model, please clearly describe them.

Author Response

In the manuscript named “CRISPR/cas9 allows for the quick improvement of tomato firmness breeding”, authors have adopted CRISPR/cas9 technology to knock out the FIS1 and PL in tomato, their results have shown the gene editing lines with improvement of firmness in tomato. These findings would be useful for tomato genetic breeding in future, but there were some comments about it.

(1) Authors have knocked two genes, but their results have not shown the detail about gene editing lines, including DNA sequencing results. The representational sequences should be displayed in in manuscript, and other results should be released by supplements, or other public method.

Response 1: Thank you for your thorough and insightful comments. The sequences of wild-type and the representational mutant lines have been shown in Fig. 1b, the other lines are present in Fig. S1.

(2) How these two genes express in gene editing lines? If there are gene expressing results present in this research, such as qRT-PCR, which would be beneficial for exploring genes’ function, and trait formation.

Response 2: We did not assess the expression levels of the mutant lines, as we identified the type of mutation in each line through PCR and Sanger sequencing. We specifically selected mutant lines that exhibited large deletions or frameshift mutations leading to premature termination of protein translation, based on our hypothesis that these types of mutations would result in the absence of any biologically functional proteins. In fact, we chose two negative regulators associated with tomato fruit firmness for gene editing; by ensuring their loss of function, we aimed to achieve our objective of enhancing fruit firmness.

(3) The results of gene editing were unclearly described, which effects were these gene editing sites? Unknown, how did these editing sites function or effect on enzyme function, authors would provide more evidences about their function.

Response 3: Thank you for pointing this out. The pertinent information regarding target design has been incorporated in Method 2.2. The gene editing description was added in Lines 178-184.

(4) The genetic background about two tomato lines, OT44 and OT45, were not well introduced in this research, why selected these two lines? They were different between some traits, such as compression resistance, soluble solids, how did two genes, FIS1 and PL, function about these traits, please describe in detail.

Response 4: The detail of two tomato lines was describe in lines 97-107.

(5) The refs were missed in many sections, such as method section, many words were selected from public methods, but refs missed. The same as discussion section, few refs were listed, and discussion needed to be described in depth.

Response 5: The reference has been added. The discussion has been revised, as shown in lines 295-300, 314-321, and 331-334.

(6) The conclusion was rather superficial and fails to be integrated with present research.

Response 6:This section has been revised.

(7) There was an excess space in line 155.

Response 7:The excess space has been deleted.

(8) Fig. S1 could not be accessed in present version, please check it.

Response 8:Figure S1 is provided in the supplementary materials.

(9) The statistical analysis was not clearly described, ANOVA, or other model, please clearly describe them.

Response 9:The statistical analysis methods have been added in lines 153-155.

Reviewer 3 Report

Comments and Suggestions for Authors

The manuscript deals with evaluation of knock out of two genes, involved in regulation of firmness in tomato, using CRISPR/Cas9. Authors have compared to obtained mutants to wild genotypes. Authors have measured several traits including firmness, color...

Te topic of the manuscript is of great interest and authors have underlined it interest for the breeding free GMO.

The manuscript is well written, the presentation is sound and the topic meets the expectation of CIMB.

Nevertheless, there are some concerns in this manuscript.

1- lack of precision. This concerns the absence of experimental design. Moreover, authors have written L93-94 "employing standard cultivation management practices"? what are standard cultivation (conventional, organic, regenerative cultivation). This lacks precision. Authors are asked to provid water fertilizers and phénological details.

2- lack of precision in the figures

For example: there are several points that are not clear in Figure 2. Are the same tomatoes used for the figures? If so, why not arrange them in the same order?
The graphics (curves), especially in the upper part (CR-fis1/pl-OT44), are not visible.
Please check all figures

3-On what basis has the colour been measured? There are scales or colour charts used internationally. It would also have been desirable to measure lycopene and/or carotenoids.

4- conclusion should be reworded carefully by presenting the most important findings, limitations of the study and perspectives.

Author Response

The manuscript deals with evaluation of knock out of two genes, involved in regulation of firmness in tomato, using CRISPR/Cas9. Authors have compared to obtained mutants to wild genotypes. Authors have measured several traits including firmness, color...

Te topic of the manuscript is of great interest and authors have underlined it interest for the breeding free GMO.

The manuscript is well written, the presentation is sound and the topic meets the expectation of CIMB.

Nevertheless, there are some concerns in this manuscript.

1- lack of precision. This concerns the absence of experimental design. Moreover, authors have written L93-94 "employing standard cultivation management practices"? what are standard cultivation (conventional, organic, regenerative cultivation). This lacks precision. Authors are asked to provide water fertilizers and phénological details.
Response 1:  Thank you for bringing this to our attention. We referenced descriptions of cultivation conditions provided by other studies; however, we did not directly quote their content. Instead, we have made modifications based on the actual cultivation conditions, as detailed in lines 104-107.

2- lack of precision in the figures

For example: there are several points that are not clear in Figure 2. Are the same tomatoes used for the figures? If so, why not arrange them in the same order?
The graphics (curves), especially in the upper part (CR-fis1/pl-OT44), are not visible. Please check all figures.

Response 2: The images in Fig. 1-5 have been upgraded to higher quality versions. They are indeed the same tomatoes in Fig. 2a, and we inadvertently mixed up the order during the weight measurements. As a result, we were unable to arrange them in sequence for the photograph.  

3-On what basis has the colour been measured? There are scales or colour charts used internationally. It would also have been desirable to measure lycopene and/or carotenoids.

Response 3: We have unfortunately lost the measurement data for colors; therefore, we have omitted the color descriptions in this revised version.

4- conclusion should be reworded carefully by presenting the most important findings, limitations of the study and perspectives.

Response 4:This section has been revised.

Reviewer 4 Report

Comments and Suggestions for Authors

The authors, Qihong Yang and co-workers, presented a manuscript entitled "CRISPR/cas9 allows for the quick improvement of tomato firmness breeding".

They used the CRISPR/Cas9 system to knock out two genes of tomato to obtain genotypes with improved firmness and storability of fruits.

Using CRISPR/Cas9 technology, they were able to obtain tomato genotypes mutated in one or both genes. They compared the obtained genotypes with the parent tomato varieties in terms of mechanical resistance of the fruit, water loss and changes in the skin of the fruit.

The mutant genotypes showed the expected changes in fruit quality in these parameters.

Other characteristics important for fruit quality, such as ripening time, size, shape and colour of the fruit, were not affected.

Their work is innovative and provides valuable insights extending recent discoveries of tomato gene function and their application to practical tomato breeding.

The submitted manuscript deserves publication in the CIMB journal. However, I request a thorough explanation of what the FIS1 gene is (line 73). On line 82, it would be useful to explain more thoroughly why these two genes were used.

Author Response

The authors, Qihong Yang and co-workers, presented a manuscript entitled "CRISPR/cas9 allows for the quick improvement of tomato firmness breeding".

They used the CRISPR/Cas9 system to knock out two genes of tomato to obtain genotypes with improved firmness and storability of fruits.

Using CRISPR/Cas9 technology, they were able to obtain tomato genotypes mutated in one or both genes. They compared the obtained genotypes with the parent tomato varieties in terms of mechanical resistance of the fruit, water loss and changes in the skin of the fruit.

The mutant genotypes showed the expected changes in fruit quality in these parameters.

Other characteristics important for fruit quality, such as ripening time, size, shape and colour of the fruit, were not affected.

Their work is innovative and provides valuable insights extending recent discoveries of tomato gene function and their application to practical tomato breeding.

The submitted manuscript deserves publication in the CIMB journal. However, I request a thorough explanation of what the FIS1 gene is (line 73). On line 82, it would be useful to explain more thoroughly why these two genes were used.

Response:  Thank you for your thorough and insightful comments. We have incorporated information regarding the regulatory role of FIS1 (Lines 81-82) in tomato firmness and elucidated the primary reasons for our selection of these two genes (Lines 91-92 and 325-337).

Round 2

Reviewer 1 Report

Comments and Suggestions for Authors

Thank you so much for your efforts 

The names of journals should be summarised 

Author Response

Comment:The names of journals should be summarised.

Response: Thank you for your dedicated efforts in enhancing the quality of our manuscript.  We referred to the MPDI guidelines regarding journal abbreviations and made the requisite adjustments to the manuscript accordingly.

Reviewer 2 Report

Comments and Suggestions for Authors

Thanks for authors work, most of my comments were well addressed in revision, but some figures were repeated in pdf file. In addition, significance could be labeled with “*” or “**” in figures. Please carefully check them. Good luck.

Author Response

Comment 1: Thanks for authors work, most of my comments were well addressed in revision, but some figures were repeated in pdf file. 

Response1: Thank you for your dedicated efforts in enhancing the quality of our manuscript. We have eliminated duplicate images and uploaded a revised PDF.

Comment 2: In addition, significance could be labeled with “*” or “**” in figures. Please carefully check them.

Response: We believe that utilizing distinct letters to represent significant differences in multiple group comparisons is more consistent with the requirements of the statistical test. Consequently, we have decided not to alter the figures.